# The Effect of the Stress State, Testing Temperature, and Hardener Composition on the Strength of an AlMg5/Epoxy Metal-Polymer Joint

**DOI:** 10.3390/ma15207326

**Published:** 2022-10-20

**Authors:** Sergey Smirnov, Dmitry Konovalov, Dmitry Vichuzhanin, Irina Veretennikova, Aleksander Pestov, Viktoria Osipova

**Affiliations:** 1Institute of Engineering Science, Ural Branch of the Russian Academy of Sciences, 620049 Ekaterinburg, Russia; 2I. Ya. Postovsky Institute of Organic Synthesis, Ural Branch of the Russian Academy of Sciences, 620049 Ekaterinburg, Russia

**Keywords:** adhesive joint, strength, temperature, Arcan specimens, Brazil-nut-sandwich specimens

## Abstract

The regularities of the effect of a complex stress state on the strength of an AlMg5/epoxy adhesive joint are experimentally studied at −50 and +23 °C in tension+shear and compression+shear tests with different normal-to-shear stress ratios. The tests use modified Arcan specimens and Brazil-nut-sandwich specimens, with the lateral faces of the adhesive layer having a shape of a mushroom-like “ridge” aimed at reducing stress concentration at the specimen edges. An original computational model of a selected microvolume including the interface together with the adjacent substrate and adhesive layers is used to process the experimental results. The attainment of the threshold value of strain energy density in the selected microvolume, *W**, is used as the failure criterion. The effect of the hardener composition, the testing temperature, and the value of the phase angle β determining the proportion of normal and shear stresses at the adhesive interface on the threshold value *W** is detected. *W**(β) diagrams (fracture loci) are plotted and analytically described logarithmic functions. They can be used to make strength calculations for adhesive joints in structures and metal-polymer composites.

## 1. Introduction

Using hybrid metal-polymer assemblies is a promising trend in developing composite materials and honeycomb and laminated load-bearing structures combining high strength and small weight [1,2,3]. Thermosetting or thermoplastic polymer adhesive materials are used as binders, providing the necessary adhesive bond.

The cohesive strength of adhesive materials, as well as the overall rigidity of structures, can be increased by using reinforcing elements, e.g., glass or carbon tapes and fabrics, hard dispersoids, etc. However, the ability of metal-polymer assemblies to resist delamination-type failure is determined exclusively by the adhesive strength of the metal–binder interface.

Cold-curing adhesives are generally used to form an adhesive interface under conditions when it is difficult or technically impossible to use heating to cure thermosetting adhesives. Like other structural materials, an adhesive used as a binder in a metal-polymer composite must provide interfacial strength sufficient for its reliable operation in a structure. This is especially important for structures designed for Arctic regions, where low environmental temperatures aggravate the effect of mechanical loads.

Composite materials and structures are currently designed mainly by means of engineering analysis systems involving global strength calculations of structural components and their assemblies, which are generally in a complex stress-strain state when being used. Reference databases included in engineering analysis software (Ansys, Abaqus, etc.) containing characteristics of the ultimate properties of structural materials lack information of polymer adhesives. Engineering documents generally contain information only on the ultimate shear (more rarely, cleavage) strength of adhesive joints and only at room temperature. Also note that there is currently no fully formed idea of what criteria should be taken to estimate the strength of adhesive joints and what characteristics of the stress state should be used for this purpose. Engineering calculations often use force criteria based on the application of conventional theories of the strength of solids, as well as criteria that were specialized for solving problems of adhesive strength (see reviews [4,5,6,7,8])). When predicting the failure of adhesive interfaces by means of force criteria one faces a number of essential limitations stemming from the impossibility of using them in boundary regions with a stress singularity and their insensitivity to the adhesive layer thickness [9]. Furthermore, the force criteria ignore the effect of the history of the stress-strain state on the failure of an adhesive interface.

An important step in predicting the failure of adhesive interfaces was the development of cohesive zone models (CZM), according to which crack propagation is preceded by the development of material damage in front of the crack tip, and crack extension occurs when damage reaches a critical level. CZMs were focused on in a special issue of the Journal of Engineering Fracture Mechanics 70 (2003) and in later reviews [10,11,12]. Energy release rate *G* is generally used as the constitutive parameter of CZMs, which is calculated as the work spent on the formation of a free surface unit during crack propagation [13]. The model also takes strength and crack-opening displacement into account. For experimental determination of *G*_c_ and identification of models by traction–separation diagrams, various tests are performed under conditions of cleavage (*G*_cI_), shear (*G*_cII_), and transverse shear (*G*_cIII_), including those regulated by ASTM D5528. The use of CZMs enables one to solve important problems of preserving the survivability of an adhesive joint in the case of the presence or formation of cracks in it. 

When an adhesive assembly is designed, it is important to determine the level of acceptable mechanical loads, above which initial discontinuities may start to appear at the interface. With further loading, these discontinuities cause the appearance and propagation of cracks and eventually the fracture of the assembly. An effective indication of this can be the ultimate value of the work/energy spent by external forces on the formation of local discontinuities at the most heavily loaded and dangerous points. The criteria based on strain energy take into account all the stress and strain components; therefore, they are more suitable as the criteria of initial fracture than stress and strain criteria. Besides, as distinct from energy release rate, this quantity is cumulative, and this provides a principal possibility to take into account the effect of the history of deformation conditions on the failure of an adhesive assembly. Strain energy density averaged over the adhesive volume being deformed was used as the failure criterion in [14]. The authors of that study performed tests only for tension of single lap joints; therefore, they did not study the dependence of threshold strain energy density at fracture on stress state characteristics.

The ultimate strain energy of an adhesive assembly depends on the stress state; however, few results of experimental studies of these dependences are found in the scientific and technical literature. The majority of them deal with *G*_c_ as dependent on the value of the phase angle ψ characterizing the ratio of shear σ_s_ and normal σ_n_ stresses in front of the crack tip [15,16,17],
ψ = atan(σ_s_/σ_n_).(1)

The representation of the threshold characteristics of adhesive assemblies in tension+shear and compression+shear tests as diagrams in the coordinates (σ_s_*~σ_n_*) is also common [18,19,20,21]. In [22,23], from the results of testing modified Arcan specimens and three-point bending of glued multilayer metal-polymer specimens, strength diagrams for an adhesive joint were plotted in the coordinates of the strain energy density components.

However, this two-parameter representation does not allow us to numerically evaluate the degree of utilization of the available strength of an adhesive joint, and this limits its design capabilities.

Thus, it follows from the foregoing brief review that the problem of the effect of the stress state on the initiation of adhesive joint failure needs to be studied in more detail. This is particularly true for relating threshold strain energy density to the parameter characterizing the stress state; this dependence can be used for evaluating the level of permissible mechanical loads under a complex stress state.

The aim of this paper is to study the effect of a complex stress state under conditions of low and room temperatures on the threshold strain energy density for adhesive joints. Since aluminum alloy adhesive joints produced with the use of a cold-curing adhesive based on epoxy resin have a promising application in metal-polymer composites and structures, they were chosen to be the object of experimental research.

## 2. Materials and Methods

The strength of adhesive joints in specimens made of the AlMg5 aluminum magnesium alloy (the Russian analog of AA5056 and AlMg5) was studied. The alloy containing 5.2% Mg, 0.5% Mn, 0.3% Si, and 0.2% Zn is used to produce structural members. Adhesive compositions were made on the basis of ED-20 epoxy (bisphenol A diglycidyl ether) (the Sverdlov Plant, Russia) with an epoxy number of 21.1%. Modified Arcan specimens [24] with glued inserts were used to study the effect of the stress state on strength under the tension+shear loading condition. The effect of the stress state under the compression+shear loading condition was studied on Brazil-nut-sandwich specimens [16,25].

Metal specimens were made from a hot-extruded bar with a diameter of 20 mm (KUMZ, Kamensk-Uralsky, Russia). The contact surfaces of the specimens were then machined by milling in order to obtain the required roughness and to remove the oxide film. A roughness Ra of 0.4 μm on the contact surfaces of the plates after machining and abrasion was determined by means of a Wyko NT1100 optical surface profiler (Veeco, Plainview, NY, USA). In order to create adhesive joints, three versions of epoxy compositions were used, differing in the use of curing agents or hardeners (Chimex Limited, St. Petersburg, Russia). The composition of the materials is shown in Table 1. The PEPA curing agent is a product of the oligomerization of ethylenimine with an average molecular weight of 200 g/mol; PAPA is a PEPA derivative acylated by higher fatty acids; DETA is an aliphatic curing agent, a dimer of ethylenimine. Curing was performed at 25 °C for 24 h. The amine content of PAPA is lower than that of PEPA; therefore, to prepare the epoxy composition A2, we used a resin-to-hardener ratio of 10:3 instead of 10:1 for A1 and A3. The latter ratio is standard for cold-curing epoxy materials. The thickness of the adhesive layer in each joint was 0.2 ± 0.02 mm. Test temperatures of −50 and +23 °C were chosen as the extreme environmental temperatures in the Arctic regions of the Euro-Asian continent.

### 2.1. Dynamic Mechanical Analysis

Investigations with an Eplexor 100 N dynamic mechanical analyzer (Netzsch Gabo Instruments, GmbH, Ahlden, Germany) were performed with the aim of determining the elastic properties of epoxy compositions, which were necessary for simulating the stress-strain state of adhesive joints during testing. 

For this purpose, cylindrical specimens with a diameter of 4 ± 0.1 mm and a height of 6 ± 0.2 mm were made by turning from 10 mm diameter bars of epoxy compositions obtained by casting into Teflon molds. The tests were conducted with the use of the time sweep software option of the device, which allows one to change the temperature in the test chamber at a constant specified static load. The temperature conditions of specimen heating/cooling were automatically controlled by the executive control systems of the device. The specimens were cooled by liquid nitrogen vapors and heated by the heating elements. In order to measure the dynamic complex elastic modulus during the tests under a static compressive load of 100 N, sinusoidal mechanical vibrations with a frequency of 1 Hz and an amplitude of 5 N were additionally applied to the specimen. The specimens were first cooled in the test chamber to a temperature of −70 °C, held for 15 min to equalize the temperature of the specimens, and then the temperature in the test chamber was raised at a rate of 1°/min until it reached +100 °C.

The parameters (static and dynamic forces, static and dynamic strains, temperature) were recorded in the function of the test time into the built-in controller of the device. The measurement results were processed by means of Eplexor 9 software of the device. The complex modulus *E** was calculated by the formula
*E** = Δσ/Δε,(2)
where Δσ and Δε are the ranges of stresses and strains in a sinusoidal loading cycle.

The complex modulus *E** is represented as a complex operator,
*E** = *E*′ + *iE*″,(3)
where *E*′ = *E** cos δ is the elastic modulus (storage modulus), *E*″ = *E** sinδ is the loss modulus, and δ is the lag angle between the strain change and the stress change under sinusoidal loading conditions (the loss angle). The real part of the complex operator (3) characterizes the elastic properties of the material, and the imaginary part characterizes the viscous ones. The higher the value tan δ = *E*″/*E*′, the greater the manifestation of the viscous properties of the material. The temperature dependences of *E** are graphically shown in Figure 1.

The values of the glass transition temperature *T*_g_ = 57 °C for A1, 59 °C for A2, and 62 °C for A3 were determined according to ASTM D7028 at the points of intersection of the tangents to the curves of the temperature dependence of $\log E^{\prime }$ before and after the curve inflections. The arithmetic mean values of the complex modulus and its components determined from testing three specimens are shown in Table 2. Since the temperatures at which the adhesive joints were mechanically tested, −50 and +23 °C, were below the glass transition temperatures, the material of the adhesives was in the glassy state during testing, and it manifested practically no viscous properties; this is demonstrated by the small values of the loss modulus *E*″ and tan(δ). 

### 2.2. Modified Arcan Specimens

Modified Arcan specimens were tested in a Zwick/Roell Z2.5 testing machine (Zwick Roell AG, Ulm, Germany) in a KTKh-20 climate chamber (NPF Tekhnologiya, St. Petersburg, Russia) keeping the test temperature constant within ±1 °C by means of a two-circuit cooler, with Freon as the refrigerant, and heat-producing electric heating elements.

In order to increase the uniformity of stresses at the contact surfaces of Arcan specimens, many researchers make the specimen edges beak-shaped [18,26,27,28,29,30], thus significantly decreasing stress concentration and the probability of the initiation of edge delamination cracks. Such specimens are disadvantageous in that they are difficult to make and hard to reuse, since to prepare the contact surface for further testing, residual adhesive often has to be removed by machining (milling, grinding), and this reduces the geometric dimensions of the contact surfaces. To avoid these problems, we proposed earlier [22] to test specimens with flat facets, but to alter the shape of the lateral surface of the adhesive layer. Finite element simulation of the testing process in the Ansys v.16.2 software was performed in the Shared Research Facilities Center of the Institute of Mathematics and Mechanics, UB RAS. The problem was solved in the elastic formulation under conditions of the plane stress state. Based on the results of computational experiments with varying the shape of the lateral surface of the adhesive layer (e.g., convex, concave, etc.), we have selected the lateral surface in the form of a mushroom-shaped “ridge” with a thickness equal to that of the adhesive layer, 0.2 mm in this case. The detailed formulation of the computational problem was presented in [22]; therefore, we here restrict ourselves to showing the simulation results. A fragment of the finite element mesh of the edge region of the Arcan specimen with the lateral surface of the adhesive layer in the form of a mushroom-like ridge is shown in Figure 2a, and the diagrams of normal and shear contact stresses, plotted according to the simulation results for the A1 adhesive composition, are shown in Figure 2b–e. The comparison of the curves for the specimen having a flat-edged adhesive layer (Figure 2b,d) with that for the specimen having a ridge-shaped lateral surface of the adhesive layer (Figure 2c,e) demonstrates that, when there is a ridge, a more favorable stress state is formed near the specimen edges than that in the specimen with a flat lateral surface of the adhesive layer; namely, the value of the normal stress concentration factor at the specimen edges in cleavage tests *k*_n_ = σ_n_/$\bar{\sigma }$_n_
*=* 0.58, and the value of the shear stress concentration factor in shear tests *k_s_* = σ_s_/$\bar{\sigma }$_s_
*=* 1.02 (here, σ_n_ and $\bar{\sigma }$_n_ are the values of the normal stresses at the specimen edge and the nominal normal stress, respectively; σ_s_ and $\bar{\sigma }$_s_ denote the same for shear stresses). Under these conditions, the appearance of edge delamination cracks is hardly probable. The existence of a ridge has an insignificant effect on the deformation force since its thickness is small. The ridge can be easily formed with a shaped scraper removing excess adhesive from the lateral surface of the inserts to be bonded. Gained in [22], the positive experience of testing specimens with a mushroom-shaped glue ridge has enabled us to use specimens of the same kind in this study.

Arcan specimens with bonded inserts (Figure 3a) were placed in the grips of the testing machine so that the angle α between the tension direction and the plane of the adhesive joint was 0, 22.5, 45, 67.5, and 90° (Figure 3b). The angle α = 90° corresponded to the scheme of cleavage testing under normal tensile stresses, and α = 0° corresponded to shear under tangential stresses. At the intermediate values of the angle α, the tension+shear complex stress state was implemented.

The specimens to be tested were fixed in the testing machine grips housed in the climate chamber. After reaching the specified temperature in the chamber, they were held for 30 min until the specimen temperature was equal to the temperature in the climate chamber. The specimens were tensioned at a rate of 1 mm/min to the fracture of the adhesive joint. The load *P* during the testing was measured with an accuracy of ±1% by means of a strain-measuring system incorporated in the testing machine unit, and the change in the dimension of the adhesive layer in the direction of loading was measured by means of an Epsilon attached extensometer with an accuracy of ±1 μm. Three Arcan specimens were tested for each test temperature and each angle α, i.e., 30 specimens for each adhesive composition.

### 2.3. Brazil-Nut-Sandwich Specimens 

For specimen preparation, 20 mm diameter bars were cut into disks with a thickness of 20 ± 0.2 mm. The disks were then cut into bars in the meridional plane and abrased to a roughness Ra of 0.4 μm on the contact surfaces, which was checked with a Veeco Wyko NT1100 non-contact profiler. The prepared halves were bonded by the epoxy compositions, with shaping the lateral surface of the adhesive layer as a mushroom-like ridge, as was described for the Arcan specimens. The appearance of the specimens and the loading condition are shown in Figure 4. When the specimens were mounted between the flat dies of the testing machine, the angle *α* between the direction of compression and the interface plane varied from 4° to 21°.

Since compression+shear testing requires a greater force than is allowed by the Zwick/Roell Z2.5 testing machine, the Brazil-nut-sandwich specimens were tested in an Instron 8801 universal testing machine (Instron, High Wycombe, UK) with a maximum load of 100 kN, in a climate chamber included in the testing machine delivery set. Cooling in the chamber was performed by passing liquid nitrogen vapors, and heating was performed by electric heat-producing elements. The temperature was automatically controlled with an accuracy of ±2 degrees by means of the standard software of the machine on the basis of the measurement data of the chromel-alumel thermocouples. All in all, 25 to 30 Brazil-nut-sandwich specimens were tested for each adhesive composition.

It follows from the simulation results [22] that the shape of the lateral surface of the adhesive layer in the form of a mushroom-shaped ridge complicates the formation and propagation of end cracks at the interface. Thus, the stress state is uniform on more than 95% of the interface length; therefore, the average values of normal σ_n_ and shear σ_s_ stresses can be used to evaluate the stress state at fracture [31],

σ_s_
*= P*/*S*·cos (α); σ_n_
*= P*/*S*·sin (α),
(4)

where *P* is the load at bond failure, and *S* is the bond area.

The attainment of the threshold value *W** of strain energy density in some selected microvolume, including the interface, is used as the criterion determining the condition for the local fracture of the adhesive joint. The selected microvolume shown in Figure 5 consists of two parts. The part of thickness *h*_2_ belongs to the substrate, and the part of thickness *h*_1_ belongs to the adhesive. It is assumed that the reinforcing filler of the adhesive layer is not in the selected microvolume, and it does not participate in the adhesive interaction with the metal constituent of the composite. This assumption seems to be fairly logical since the adhesive interaction occurs at the interface between the adhesive and the substrate. The microvolume is in equilibrium, and on its opposite faces there are equal stresses σ_ij_ in the local coordinate system (*x*’, *y*’, *z*’). The *z*’-axis is directed along the normal to the interface, and the *x*’- and *y*’-axes lie in its plane.

The stresses σ_ij_ and the corresponding strains ε_ij_ are determined from solving the problem of mechanics on finding the stress-strain state of bonded solids under external loading. In the case under study, the selected microvolume is considered to be in the plane stress state under the action of normal and shear stresses determined by Equation (4). Since the loading diagrams of the specimens being tested were straight lines up to fracture, it was assumed that fracture occurs within the elastic strain of the adhesive.

The threshold strain energy density in the selected microvolume *W** and its component at the fracture of the joint is determined as
(5)$$W*\;=\;W_{n}*\;+\;W_{s}*,\;W_{n}*\;=\;({\sigma_{x’}}^{2}\;-\;\bar{\nu }\;\sigma_{x’}\sigma_{z’}\;+\;{\sigma_{z’}}^{2})/(2\bar{E}),\;W_{s}*\;=\;{\sigma_{s}}^{2}\;/(2\bar{G}),\;\bar{G}\;=\;\bar{E}/(2(1\;+\;\bar{\nu })),$$
where *W*_n_*, *W*_s_* are the normal and shear components of strain energy density; $\bar{E}$, $\bar{G}$, $\bar{\nu }$ are the effective values of the normal elastic modulus, the shear modulus, and Poisson’s ratio.

The effective values of the elastic characteristics can be calculated by the known formulas for unidirectional composites [31,32],
(6)$$\frac{h}{\bar{E}}=\frac{h_{1}}{E_{1}}+\frac{h_{2}}{E_{2}},\;\bar{\nu }=\frac{\nu_{1}h_{1}E_{2}+\nu_{2}h_{2}E_{1}}{h_{1}E_{2}+h_{2}E_{1}},\;\frac{h}{\bar{G}}=\frac{h_{1}}{G_{1}}+\frac{h_{2}}{G_{2}},$$
where *E*_1_, ν_1_, *G*_1_ and *E*_2_, ν_2_, *G*_2_ are the values of the normal elastic modulus, Poisson’s coefficient, and shear modulus of the binder and the substrate, respectively. In the calculations, the values of the elastic characteristics *E*_2_ = 71 GPa, ν_2_ = 0.3, *G*_2_ = 27 GPa for AlMg5 were set according to the reference data [33].

The formula represented by Equation (6) cannot be used directly for calculations since the values of the layer thicknesses *h*_1_ and *h*_2_ in the selected microvolume are introduced mathematical abstractions; therefore, they are physically undefined. It is only their ratio that can be physically meaningful. It characterizes the involvement of both materials in adhesive interaction. Suppose, in the first approximation, that this ratio is inversely proportional to the values of the normal elastic moduli of the adhesive and the substrate, i.e.,
(7)$$h_{1}=h_{2}\;\frac{E_{2}}{E_{1}}.$$

Then, simple transformations of Equation (6) result in the following formulas, without *h*_1_ and *h*_2_, for calculating the effective values of $\bar{E}\;$, $\bar{G}$, and $\bar{\nu }$ in the selected microvolume:(8)$$\bar{E}=\frac{E_{1}+E_{2}}{\frac{E_{1}}{E_{2}}+\frac{E_{2}}{E_{1}}}\;,\;\bar{\nu }=\frac{\nu_{1}E_{2}^{2}+\nu_{2}E_{1}^{2}}{E_{1}^{2}+E_{2}^{2}},\;\bar{G}=\frac{G_{1}+G_{2}}{\frac{G_{1}}{G_{2}}+\frac{G_{2}}{G_{1}}}\cdot$$

## 3. Results and Analysis

It follows from dynamic mechanical analysis that the material of the adhesives was in the glassy state during the tests; therefore, during deformation in the elastic region, *E** can be considered equal to normal Young’s elastic modulus, i.e., *E*_1_ = *E**. Using the data from Table 2, we have calculated the effective characteristics of the elastic properties of the selected microvolumes of adhesive joints. The values are presented in Table 3, from which it follows that the values of the effective elastic moduli are 3.9 to 6.8% higher than those of the material of the epoxy compositions and that the difference in the Poisson coefficient values is at most 0.1%.

To make a quantitative evaluation of the effect of the stress state on the strength of the adhesive joint, we use the dependence of *W** on the phase angle β calculated by the formula
(9)$$\beta =\;\mathrm{atan}\left(\frac{\sigma_{n}}{\sigma_{S}}\right).$$

Figure 6 shows the fracture loci $W^{*}\left(\beta \right)$ for the adhesive joints under study (the dots in the figures represent the experimental data averaged from three to five tests). The value of threshold strain energy density at fracture *W** is seen to increase as the values of the phase angle β (taking into account the sign) decreases. This dependence reflects the effect of normal stresses on the cleavage failure of adhesive joints as follows: when β > 0, the interface is affected by external tensile normal stresses inducing cleavage; when β < 0, there are compressive normal stresses hindering cleavage; β = 0 corresponds to the stress state of simple shear, when there is no action of normal stresses. For all the joints, *W** is higher under shear than under cleavage. Note that, as distinct from the use of the phase angle calculated by Equation (1), the fracture loci have no discontinuities when β is used to characterize the stress state. The rate of increase of *W** is much higher in the region of compressive normal stresses (β < 0) than in the region of tensile normal stresses (β > 0). The comparison of the obtained diagrams shows that, at room temperature, the adhesive joint with A2 (PAPA hardener) has the highest fracture energy consumption (the highest value of *W**) throughout the stress range studied, and the adhesive joint with A1 (PEPA hardener) has the lowest one. The situation is indefinite at −50 °C, namely: when β > 0, the values of *W** are a little higher for the joint with A2, but when β < 0, the threshold strain energy density of the joint with A3 (DETA hardener) rapidly increases; therefore, using DETA as the hardener for adhesive assemblies operating at low temperatures is preferable.

To make the estimate of the effect of test temperature on *W**(β) for each adhesive joint more obvious, let us study the curves in Figure 7. It follows from Figure 7a that *W** for the joint with A2 at −50 °C is higher when β > 0 than at +23 °C. Under shear (β = 0), the values of *W** are identical at both temperatures, and at β < 0 and *T* = −50 °C, the values of *W** are lower than at +23 °C. For the joint with A3, the trend of the effect of test temperature depending on the stress state is opposite to that for A2; namely, *W** is higher at +23 °C when β > 0, and it is lower at −50 °C when β < 0. For the joint with A1, the values of *W** are higher at −50 °C in the entire range of β.

The dependence *W**(β) can be used as a criterion for the nucleation of cleavage fracture of adhesive joints in structures and metal-polymer composites. In this case, the used strength of the joint Ψ and its residual strength margin Ψ_res_ can be quantitatively evaluated as follows:(10)$$\Psi =\frac{W}{W^{*}\left(\beta \right)},\;\Psi_{\mathrm{res}}=1-\frac{W}{W^{*}\left(\beta \right)},$$
where *W* is the local strain energy density in the most stressed (dangerous) place of the adhesive joint at a given time.

The parameters Ψ and Ψ_res_ can be conveniently used in design and checking calculations of adhesive assemblies. Note that the selection of the form of assumption (7) will have no effect on the values of Ψ and Ψ_res_ provided that the effective elastic characteristics of the selected microvolumes will be determined with the use of the same versions of assumption (7) in the determination of the experimental dependence *W**(β) and in the calculation of *W* values for the same composition used in the structural component.

The dependences *W**(β) are approximated by the following function:log *W**(β) = *c*_1_
*exp*(−*c*_2_ β) + *c*_3_, (11)
where *c*_1_, *c*_2_, *c*_3_ are empiric coefficients, whose values for the adhesive assemblies under study are shown in Table 4. 

The approximation was made in Statistica v.8 software by experimental data processing with the application of the least square method and the quasi-Newton procedure. It follows from Table 4 that the arithmetic mean error Δ in the description of the experimental data by Equation (11) is at most 7.3%.

## 4. Conclusions

The regularities of the effect of the stress state type under conditions of low and room temperatures on threshold strain energy density for AlMg5/epoxy adhesive joints with polyethylenepolyamine, polyamidepolyamine, and diethylenetriamine curing agents have been studied in experiments on the tension of modified Arcan specimens (tension+shear) and compression of Brazil-nut-sandwich specimens (compression+shear). 

It has been proposed that the attainment of the threshold value of strain energy density in a selected microvolume, *W**, should be used as the failure criterion for adhesive joints. This value depends on the test temperature and the magnitude of the phase angle β determining the normal-to-shear stress ratio at the adhesive bond interface. A computational model of a selected microvolume including the interface together with the adjacent substrate and adhesive layers has been used to process the experimental results. The dependence *W**(β) has been described by an exponential equation, and the numerical values of the approximation coefficients have been found.

The analysis of the fracture loci *W**(β) yields the following conclusion: at a temperature of +23 °C, the adhesive joint with the hardener A2 (polyamidepolyamine) has the highest *W** in the whole range of the stress state; at a test temperature of −50 °C, in the region of compressive normal stresses (β < 0), the highest strength is shown by the adhesive joint with the hardener A2, while the joint with the hardener A3 (diethylenetriamine) has the highest strength in the region of tensile normal stresses (β > 0).

The results of statistical analysis have enabled us to describe the dependence *W**(β) by an exponential equation and to determine the numerical values of the approximation coefficients. The obtained dependences can be used for calculating the strength of adhesive joints in structures and metal-polymer composites.

## Figures and Tables

**Figure 1 materials-15-07326-f001:**
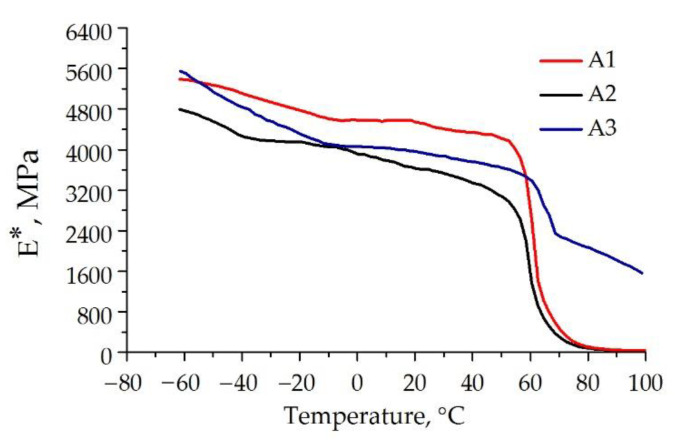
Effect of temperature on the behavior of *E** for the materials of the epoxy compositions.

**Figure 2 materials-15-07326-f002:**
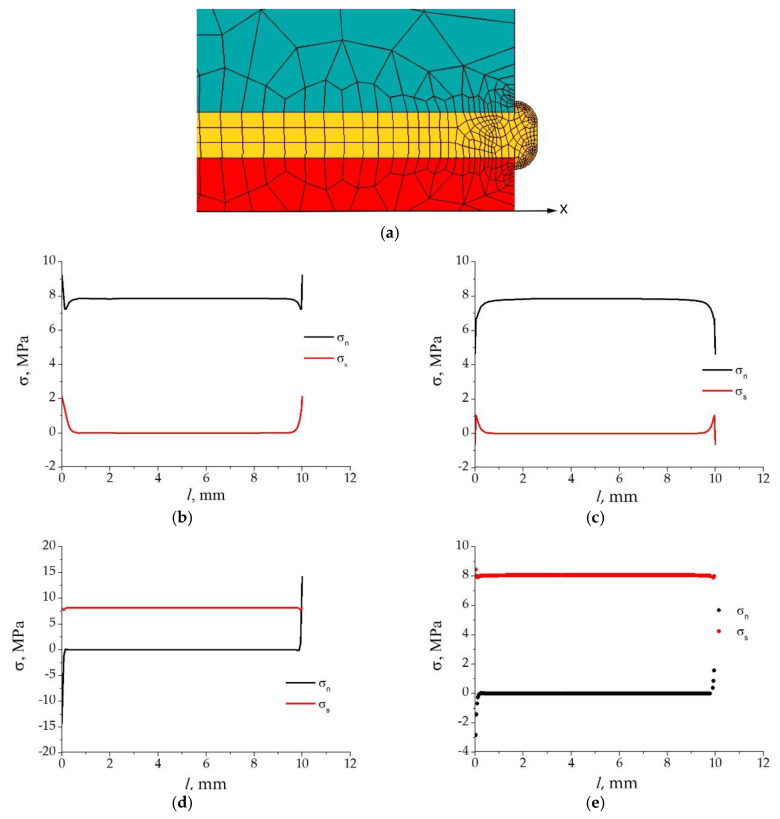
A fragment of the finite element mesh of the edge region of the Arcan specimen (**a**); the diagrams of normal and shear contact stresses for the specimen with a flat-edged adhesive layer (**b**,**d**) and that with a ridge-shaped lateral surface of the adhesive layer (**c**,**e**) in cleavage (**b**,**c**) and shear (**d**,**e**) testing.

**Figure 3 materials-15-07326-f003:**
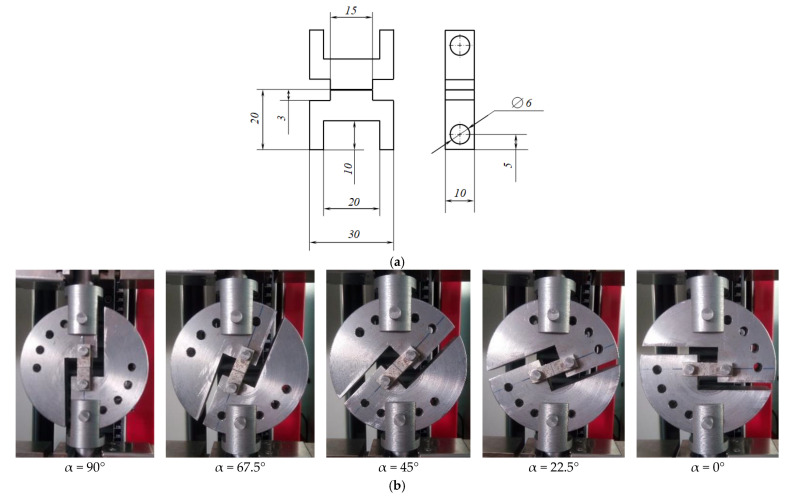
Modified Arcan specimens: an insert from the bonded assembly, dimensioned in mm (**a**); the specimens in the testing machine grips, with different orientation relative to the tensile axis (**b**).

**Figure 4 materials-15-07326-f004:**
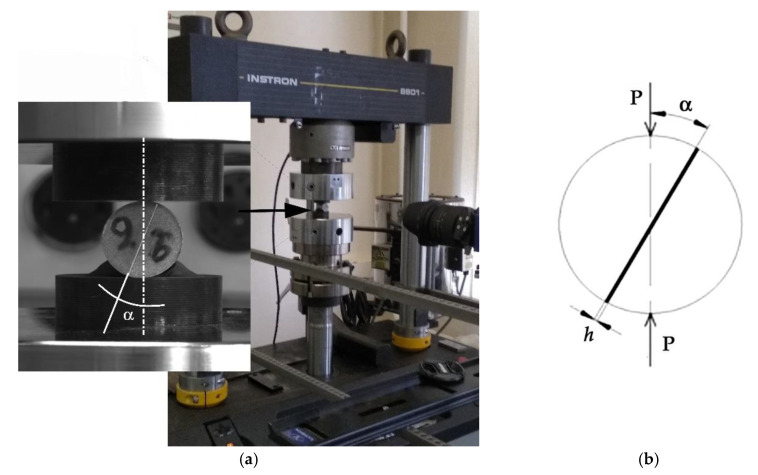
The Brazil-nut-sandwich mounted in the flat dies of the testing machine (**a**); load application pattern (**b**).

**Figure 5 materials-15-07326-f005:**
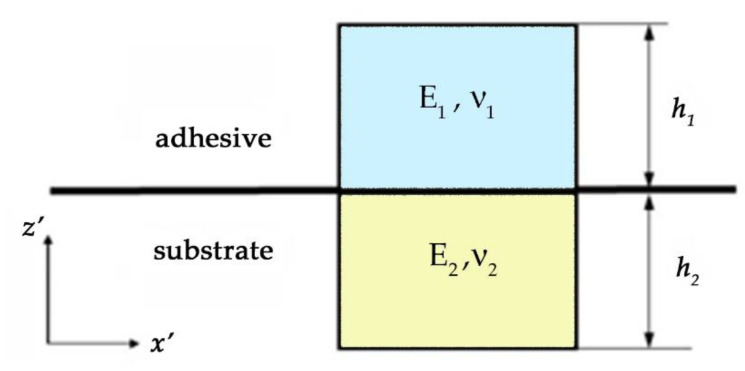
The selected microvolume at the interface.

**Figure 6 materials-15-07326-f006:**
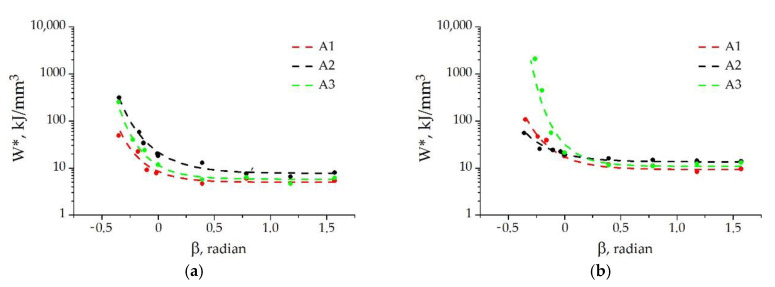
The fracture loci *W**(β) for the adhesive joints at test temperatures of +23 °C (**a**) and −50 °C (**b**): dots—experimental data; dashed lines—approximation by Equation (9).

**Figure 7 materials-15-07326-f007:**
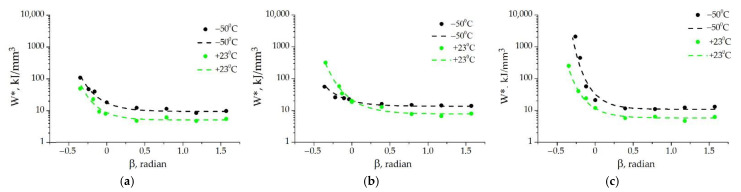
The fracture loci *W**(β) of the adhesive joints with A1 (**a**), A2 (**b**), and A3 (**c**): dots—experimental data, dashed lines—approximation by Equation (9).

**Table 1 materials-15-07326-t001:** The constitution of the epoxy compositions.

Epoxy Composition Symbol	Hardener	Resin-to-Hardener Ratio
A1	Polyethylenepolyamine (PEPA)	10:1
A2	Polyamidepolyamine (PAPA)	10:3
A3	Diethylenetriamine (DETA)	10:1

**Table 2 materials-15-07326-t002:** The values of the complex module and its components for the adhesive compositions at the test temperatures.

Epoxy Composition Symbol	*T*, °C	*E*′, MPa	*E*″, MPa	|*E**|, MPa	tan δ
A1	−50	5195 ± 15	163 ± 5	5240 ± 14	0.0314 ± 0.0011
	+23	4434 ± 12	41 ± 3	4519 ± 12	0.0092 ± 0.0012
A2	−50	4582 ± 16	118 ± 7	4591 ± 15	0.0258 ± 0.0018
	+23	3609 ± 8	65 ± 2	3609 ± 8	0.0180 ± 0.0006
A3	−50	5109 ± 14	255 ± 6	5115 ± 14	0.0499 ± 0.0013
	+23	3943 ± 10	26 ± 5	3944 ± 10	0.0066 ± 0.0003

**Table 3 materials-15-07326-t003:** The effective characteristics of the elastic properties of the selected microvolume.

Epoxy Composition Used in the Adhesive Joint	*T*, °C	$\bar{E}$, GPa	$\frac{\bar{E}-E}{E}100\%$	$\bar{G}\;$, GPa	$\frac{\bar{G}-G}{G}100\%$
A1	−50	5.6	6.8	2.1	6.4
	+23	4.7	5.9	1.7	5.3
A2	−50	4.9	5.7	1.8	5.7
	+23	3.8	4.8	1.4	4.6
A3	−50	5.4	6.7	2.0	6.3
	+23	3.9	5.3	1.5	5.0

**Table 4 materials-15-07326-t004:** Approximation coefficients in Equation (11).

Epoxy Composition Used in the Adhesive Joint	*T*, °C	Empiric Coefficients in Equation (11)			Δ, %
		*c* _1_	*c* _2_	*c* _3_	
A1	−50	0.228	−4.619	0.699	−7.3
	+23	0.271	−4.080	0.967	3.1
A2	−50	0.450	−3.636	0.887	−3.7
	+23	0.163	−3.834	1.133	−0.9
A3	−50	0.293	−4.743	0.763	2.2
	+23	0.467	−5.228	1.036	−4.4

## Data Availability

Not applicable.

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
