# Peer review of "The Effect of the Stress State, Testing Temperature, and Hardener Composition on the Strength of an AlMg5/Epoxy Metal-Polymer Joint"

_materials, 2022, doi:10.3390/ma15207326_

Round 1
Reviewer 1 Report
The manuscript is well-written and easy to follow. The description of the experiments for measuring the mechanical properties is very interesting for overcoming the challenges of bonding strength of adhesives in different environments. I suggest a major revision for the present work in some points below.
1. Mistyping of “Abaqus” in line 47
2. The introduction needs some improvement: lines 65 – 67, the CZM model depends on the energy released rate G, strength and crack-opening displacement, not only G! From the penultimate paragraph (lines 90-94), you jump immediately to the aim of the work with any explanation. This does not make sense. You should explain why your work is relevant to the models you mentioned before. Otherwise, why did you mention the models? Also, why is your choice for the materials used? You need to develop this more.
3. In Table 2 there is no error, it is not reproducible if you keep it like that.
4. Lines 185-191, You perform essential simulation for the design of the experimental setup. You must show the results here. Even in your conclusion (line 362), you mentioned the simulation results.
5. Please state how many samples were tested in the material and methods.
6. Eq. (4) is a simple equation from textbooks. Where is the reference there? Also, I think it should be in the material and methods, not in the result section.
7. What are your criteria to choose the microvolume? You should show a convergence study because otherwise, one can fit your experimental data with any equation.
8. Conclusion is a list of points from each section of the work. Not the proper conclusion of the work. What is the main outcome of your work? Connect it with the problem you state in the introduction.
Author Response
We would like to thank the Reviewer for a careful review of our manuscript and for providing us with comments and suggestions on the improvement of its quality. We have answered each of your points below. The changes to the manuscript are shown by using a green highlighter pen in MS Word.
The writing of the manuscript has been rechecked and corrected.
Comment 1. Mistyping of “Abaqus” in line 47
Response: Thanks. This has been corrected.
Comment 2. The introduction needs some improvement: lines 65 – 67, the CZM model depends on the energy released rate G, strength and crack-opening displacement, not only G! From the penultimate paragraph (lines 90-94), you jump immediately to the aim of the work with any explanation. This does not make sense. You should explain why your work is relevant to the models you mentioned before. Otherwise, why did you mention the models? Also, why is your choice for the materials used? You need to develop this more.
Response: Thanks. We have added the new information to Introduction.
Comment 3. In Table 2 there is no error, it is not reproducible if you keep it like that.
Response: Thanks. This has been corrected.
Comment 4. Lines 185-191, You perform essential simulation for the design of the experimental setup. You must show the results here. Even in your conclusion (line 362), you mentioned the simulation results.
Response: Thanks. We have added the new information to text.
Comment 5. Please state how many samples were tested in the material and methods.
Response: Thanks. We have added the new information to text.
Comment 6. Eq. (4) is a simple equation from textbooks. Where is the reference there? Also, I think it should be in the material and methods, not in the result section.
Response: Thanks. This has been corrected.
Comment 7. What are your criteria to choose the microvolume? You should show a convergence study because otherwise, one can fit your experimental data with any equation.
Response: Thanks. We have added the new information to text.
Comment 8. Conclusion is a list of points from each section of the work. Not the proper conclusion of the work. What is the main outcome of your work? Connect it with the problem you state in the introduction.
Response: Thanks. We have modified it.
Please see the attachment article.

Reviewer 2 Report
The strength of the AlMg5/Epoxy metal-polymer joint was studied through analyzing the effect of the stress state, testing temperature and hardener composition. The research is very interesting. The manuscript could be accepted for publication, but some changes still should be made.
1. Please explain how to choose and decide the hardener and the resin-to-hardener ratio in Table 1.
2. Please change the color of A3 in Fig. 5 in order to improve identifiability.
3. The contents of the conclusions need to be concise.
4. The key result of this study shown in abstract was too low. Some important law, phenomenon or reason should be mentioned in the abstract.
Author Response
We would like to thank the Reviewer for a careful review of our manuscript and for providing us with comments and suggestions on the improvement of its quality. We have answered each of your points below. Changes to the manuscript are shown by using a green highlighter pen in MS Word.
Comment 1. Please explain how to choose and decide the hardener and the resin-to-hardener ratio in Table 1
Response: Thanks. The resin-to-hardener ratio of 10:1 for polyethylenepolyamine (PEPA) and diethylenetriamine (DETA) is standard for cold curing epoxy. The amine content in PAPA is lower than that in PEPA; therefore, we used a resin-to-hardener ratio of 10:3 instead of 10:1. The suggested correction has been made.
Comment 2. Please change the color of A3 in Fig. 5 in order to improve identifiability.
Response: Thanks. We have modified it.
Comment 3. The contents of the conclusions need to be concise.
Response: Thanks. We have modified it.
Comment 4. The key result of this study shown in abstract was too low. Some important law, phenomenon or reason should be mentioned in the abstract.
Response: Thanks. We have modified it.

Round 2
Reviewer 1 Report
The authors have corrected all points mentioned in the major review, which certainly improved the work. The conclusion doe not yet answer the main general problem and there still is room for improvements.